# Association between serum uric acid and left ventricular hypertrophy/left ventricular diastolic dysfunction in patients with chronic kidney disease

Il Young Kim[1,2], Byung Min Ye[1,2], Min Jeong Kim[1,2], Seo Rin Kim[1,2], Dong Won Lee[1,2], Hyo Jin Kim[1,3], Harin Rhee[1,3], Sang Heon Song[1,3], Eun Young Seong[1,3], Soo Bong Lee[1,2]*

1 Department of Internal Medicine, Pusan National University School of Medicine, Yangsan, Republic of Korea, 2 Research Institute for Convergence of Biomedical Science and Technology, Pusan National University Yangsan Hospital, Yangsan, Republic of Korea, 3 Medical Research Institute, Pusan National University Hospital, Busan, Republic of Korea

* sbleemd@pusan.ac.kr

**Data Availability Statement:** All relevant data are within the manuscript.

## Abstract

### Background

The level of serum uric acid (SUA) has been reported to be associated with left ventricular hypertrophy (LVH) and left ventricular diastolic dysfunction (LVDD). However, this association remains unclear in patients with chronic kidney disease (CKD).

### Methods

A total of 1025 patients with pre-dialysis CKD with preserved left ventricular systolic function were enrolled in this cross-sectional study. The LVH and LVDD were assessed using two-dimensional echocardiography and tissue Doppler imaging. The associations of LVH/LVDD with clinical and laboratory variables were investigated using univariable and multivariable logistic regression analyses.

### Results

In a multivariable analysis, the SUA level was an independent predictor of LVH (odds ratio [OR]: 1.40, 95% confidence interval [CI]: 1.31–1.50, P < 0.001). In addition, patient age, systolic blood pressure, intact parathyroid hormone levels, and left atrial volume index levels were independent predictors of LVH. The SUA level was also an independent predictor of LVDD (OR: 1.93, 95% CI: 1.53–2.43, P < 0.001). Furthermore, systolic blood pressure and left atrial volume index levels were an independent predictor of LVDD. Receiver-operating characteristic curve analysis showed that the best cutoff values of SUA levels for identifying LVH and LVDD were ≥ 7.5 mg/dL and ≥ 6.3 mg/dL, respectively.

**Funding:** The author(s) received no specific funding for this work.

**Competing interests:** The authors have declared that no competing interests exist.

## Conclusion

The SUA level was an independent predictor of LVD and LVDD in patients with CKD, suggesting that SUA could be a biomarker for LVH and LVDD.

## Introduction

Cardiovascular disease (CVD) is the primary cause of death in patients with chronic kidney disease (CKD) [1]. Patients with CKD not only have a high burden of traditional risk factors for CVD, but also have CKD-related risk factors such as inflammation, increased levels of calcium and phosphorus products, uremic toxins, anemia, and fluid overload [2]. The cardiovascular system is closely related to renal function, as renal dysfunction can contribute to the heart's structural and functional abnormalities, which can worsen renal function [3]. Of all cardiac problems in patients with CKD, left ventricular hypertrophy (LVH) and left ventricular diastolic dysfunction (LVDD) are common and closely related to increased CVD mortality in these patients [2, 4]. Accordingly, the verification of predictors of LVH and LVDD is essential in CVD risk stratification of patients with CKD.

Uric acid is the end-product of purine metabolism in humans, and hyperuricemia is common in patients with CKD due to decreased uric acid clearance [5, 6]. Beyond its role in gout, previous epidemiologic studies have suggested that an increased serum uric acid (SUA) level is a risk factor for various cardiovascular conditions, including hypertension, metabolic syndrome, coronary artery disease, cerebrovascular disease, vascular dementia, and kidney disease [7]. Additionally, elevated SUA levels are reported to be associated with LVH and LVDD [8–11]. However, the association between SUA levels and LVH/LVDD is not well known in the CKD population.

In this study, taking into account the high prevalence of hyperuricemia, LVH, and LVDD in patients with CKD, we hypothesized that an increased SUA level is a risk factor for LVH and LVDD in these patients. To verify this hypothesis, we investigated the associations between SUA levels and LVH/LVDD measured by echocardiography in patients with pre-dialysis CKD.

## Materials and methods

### Study population

In this cross-sectional study, we retrospectively reviewed adult patients ($\geq$ 18 years old) who had visited the nephrology clinic in Pusan National University Yangsan Hospital between 2010 and 2018. The estimated glomerular filtration rate (eGFR) was determined by the Modification of Diet in Renal Disease equation [12]: $186 \times$ serum creatinine levels$^{-1.154} \times$ patient age$^{-0.203} \times 0.742$ (if female) or $\times 1.21$ (if African-American). All study subjects had CKD (eGFR < 60 mL/min/1.73 m$^2$) and were not on dialysis. Depending on the eGFR value, each patient was classified into one of the 3 CKD groups: CKD stage 3 (n = 511), $30 \leq$ eGFR < 60 mL/min/1.73 m$^2$; CKD stage 4 (n = 356), $15 \leq$ eGFR <30 mL/min/1.73 m$^2$; CKD stage 5 (n = 158), eGFR < 15 mL/min/1.73 m$^2$. To exclude patients with acute kidney injury and to identify patients with CKD, only patients whose previous serum creatinine levels were known from medical records or who were followed for at least 3 months were included. Patients with valvular heart disease, congenital heart disease, cardiomyopathy, evidence of systolic heart failure (ejection fraction < 50%), and atrial fibrillation were excluded because these abnormalities

could potentially confound the relationship between SUA levels and LVH/LVDD. The study protocol was approved by the Institutional Review Board of Pusan National University Yangsan Hospital (IRB No. 05-2018-118). All research and data collection processes were conducted in accordance with the Declaration of Helsinki and current ethical guidelines. The Institutional Review Board of Pusan National University Yangsan Hospital waived the need for informed consent due to the retrospective nature of the analysis that only used the information available from anonymized medical charts and records.

## Study variables

Demographic and clinical data including patient age, sex, diabetes, gout, history of cardiovascular disease (coronary heart disease, cerebrovascular disease, peripheral vascular disease), concurrent medication [angiotensin-converting enzyme inhibitor (ACEI), angiotensin receptor blocker (ARB), calcium channel blocker, beta-blocker, thiazide/loop diuretics, urate-lowering agent], body mass index (BMI), and blood pressure were obtained by reviewing the medical records of patients. Diabetes was defined as a fasting plasma glucose concentration of $\geq$ 126 mg/dL or a hemoglobin A1c percentage of $\geq$ 6.5%. Blood pressure was measured from each patient's upper right arm in a sedentary position using an automated sphygmomanometer after a 5-min rest. BMI was calculated by measuring each patient's weight and height and was expressed as kg/m$^2$. All blood variables, including levels of SUA, albumin, calcium, phosphate, total cholesterol, hemoglobin, C-reactive protein (CRP), and intact parathyroid hormone (PTH), were measured concomitantly. The amount of urinary albumin was measured by calculating the urinary albumin to creatinine ratio (mg/g Cr).

## Echocardiography

All study subjects had undergone transthoracic echocardiography using an IE33 echo system (Philips, Amsterdam, The Netherlands), based on previous reports. All echocardiographic data was performed according to the guideline of the American Society of Echocardiography [13] and were analyzed by an experienced cardiologist who was blinded to clinical details. Briefly, using the M-mode in the parasternal long-axis view, left ventricular (LV) mass was estimated by the cube formula at end-diastole (LV mass = 0.8 × [1.04 × {interventricular septum thickness + LV internal diameter + posterior wall thickness}$^3$ – {LV internal diameter}$^3$] + 0.6 g). LV mass index (LVMI) was calculated by dividing the LV mass by the patient's body surface area (BSA) [LVMI = LV mass (g)/BSA (m$^2$)] [13]. LVH was defined as LVMI > 115 g/m$^2$ in men and > 95 g/m$^2$ in women [13]. The left ventricular ejection fraction (LVEF), which indicates LV systolic function, was calculated using the biplane Simpson's method. Diastolic dysfunction was assessed using both Doppler echocardiography and tissue Doppler imaging. Early mitral inflow velocity (E) and late mitral inflow velocity (A) were measured using Doppler echocardiography [13]. Peak early mitral annular velocity (e') was computed as the average of velocities obtained at the medial and lateral annuli using tissue Doppler [13]. The E/e' ratio was calculated and used for the estimation of LV filling pressure. The severity of diastolic dysfunction was assessed using the e' values and E/e' ratios [13], according to guideline of the American Society of Echocardiography for the evaluation of left ventricular diastolic dysfunction [14]. Left atrial volume index, E/A, deceleration time of E, e', and E/e' was used to categorize diastolic dysfunction into normal function, or grades 1, 2, or 3 diastolic dysfunction. The presence of LVDD was defined as $\geq$ grade 1 dysfunction.

## Statistical analysis

Continuous variables are expressed as mean ± standard deviation, while categorical variables are presented as percentages. Differences among groups were tested with one-way analysis of variance for continuous variables and the chi-square test for categorical data. Pearson's correlation was used to investigate the correlation between SUA levels and echocardiographic findings. Univariable and multivariable logistic regression analyses were performed to calculate the odds ratio (OR) with a 95% confidence interval (CI) for predicting LVH and LVDD. Significant variables were identified by univariable analysis, and the clinically important variables were selected for multivariable analysis. Receiver-operating characteristic (ROC) curve analysis was performed to assess the area under the curve (AUC) and Youden index was used to determine the best cutoff value of SUA levels for predicting LVH and LVDD in study subjects. To assess the AUC for the combine factors, logistic regression was applied to calculate the predictive probability of combined factors. ROC curves were constructed using the predictive probability as a covariate. AUCs were compared using the method described by Delong et al. [15]. A value of $P < 0.05$ was considered statistically significant. All analyses were performed using the SPSS version 26.0 statistical package (SPSS, Inc., Chicago, IL, USA) and MedCalc Statistical Software version 19.4.1 (MedCalc Software, Ostend, Belgium).

## Results

### Baseline characteristics of study population

The baseline characteristics of the study population according to CKD stage are shown in Table 1. Of the 1025 patients, 511 were in CKD stage 3, 356 in CKD stage 4, and 158 in CKD stage 5. The mean eGFRs (mL/min/1.73 m$^2$) were 42.8 ± 8.5 in CKD stage 3, 22.2 ± 4.2 in CKD stage 4, and 9.9 ± 3.4 in CKD stage 5. There were no significant differences across the three groups in terms of sex, prevalence of diabetes, BMI, diastolic blood pressure, and total cholesterol levels. Patients with higher CKD stages were more likely to be old ($P < 0.001$); have cardiovascular diseases (coronary heart disease [$P = 0.010$], cerebrovascular disease [$P < 0.001$], peripheral vascular disease [$P = 0.002$]), and gout ($P = 0.020$)]; receive anti-hypertensive medication [ACEI or ARB ($P = 0.033$), calcium channel blockers ($P = 0.009$), beta blockers ($P < 0.001$), thiazide diuretics ($P < 0.001$), loop diuretics ($P < 0.001$), and urate-lowering agents ($P < 0.001$)]; have elevated systolic blood pressure ($P < 0.001$), and elevated levels of urinary albumin ($P < 0.001$), SUA ($P = 0.004$), phosphate ($P < 0.001$), CRP ($P < 0.001$), intact PTH ($P < 0.001$), and left atrial volume index ($P < 0.001$); and have decreased levels of serum albumin ($P < 0.001$), calcium ($P < 0.001$), and hemoglobin ($P < 0.001$). Among the echocardiographic parameters, patients with higher CKD stages had a higher LVMI ($P < 0.001$) and prevalence of LVH (29.9% in CKD stage 3, 46.9% in CKD stage 4, and 66.5% in CKD stage 5, $P < 0.001$). The degree of LVDD was more severe with increasing CKD stages, as evidenced by a lower e' ($P < 0.001$) and higher E/e' ratio ($P < 0.001$). Patients with higher CKD stages had higher values of left atrial volume index ($P = 0.014$). However, there were no significant differences between the three CKD groups in terms of the prevalence of LVDD.

The baseline characteristics of the study population according to tertiles of SUA are shown in Table 2. Patients with a higher tertile of SUA were older ($P = 0.001$) and were more likely to have higher levels of systolic blood pressure ($P < 0.001$), urinary albumin ($P = 0.001$), phosphate ($P < 0.001$), CRP ($P = 0.031$), and intact PTH ($P < 0.001$) and lower levels of eGFR ($P < 0.001$), albumin ($P = 0.007$), and hemoglobin ($P < 0.001$). Among the echocardiographic parameters, patients with a higher tertile of SUA tended to have higher values of LVMI ($P < 0.001$), left atrial volume index ($P < 0.001$), and E/e'

**Table 1. Baseline characteristics of the study population according to CKD stage (n = 1025).**

| | CKD stage 3 | CKD stage 4 | CKD stage 5 | P[c] |
| --- | --- | --- | --- | --- |
| | (n = 511) | (n = 356) | (n = 158) | |
| Age (years) | 58.6 ± 9.9 | 61.2 ± 10.2 | 63.3 ± 12.0 | <0.001 |
| Sex, male [n (%)] | 262 (51.3%) | 189 (53.1%) | 88 (55.7%) | 0.841 |
| Diabetes [n (%)] | 259 (50.7%) | 178 (50.0%) | 85 (53.8%) | 0.721 |
| Cardiovascular disease [n (%)] | | | | |
| Coronary heart disease[a] | 88 (17.2%) | 83 (23.3%) | 43 (27.2%) | 0.010 |
| Cerebrovascular disease[b] | 40 (7.8%) | 41 (11.5%) | 31 (19.6%) | <0.001 |
| Peripheral vascular disease | 28 (5.5%) | 30 (8.4%) | 22 (13.9%) | 0.002 |
| Medication [n (%)] | | | | |
| ACEI or ARB | 373 (73.0%) | 274 (77.0%) | 131 (82.9%) | 0.033 |
| Calcium channel blocker | 298 (58.3%) | 236 (66.3%) | 110 (69.6%) | 0.009 |
| eta blocker | 157 (30.7%) | 149 (41.9%) | 95 (60.1%) | <0.001 |
| Diuretics (thiazide) | 194 (38.0%) | 74 (20.8%) | 19 (12.0%) | <0.001 |
| Diuretics (loop) | 186 (36.4%) | 166 (46.6%) | 102 (64.6%) | <0.001 |
| Urate-lowering therapy | 75 (14.7%) | 76 (21.3%) | 47 (29.7%) | <0.001 |
| Gout | 67 (13.1%) | 60 (16.9%) | 35 (22.2%) | 0.020 |
| Body mass index (kg/m$^2$) | 23.5 ± 2.5 | 23.6 ± 2.5 | 23.4 ± 2.0 | 0.716 |
| Systolic blood pressure (mmHg) | 129.6 ± 18.4 | 135.3 ± 18.4 | 142.3 ± 16.2 | <0.001 |
| Diastolic blood pressure (mmHg) | 79.7 ± 14.4 | 80.3 ± 14.5 | 81.3 ± 15.2 | 0.438 |
| eGFR (mL/min/1.73 m$^2$) | 42.8 ± 8.5 | 22.2 ± 4.2 | 9.9 ± 3.4 | <0.001 |
| Urinary albumin (mg/g Cr) | 812.1 ± 818.2 | 1355.3 ± 1163.7 | 2241.0 ± 1421.9 | <0.001 |
| Albumin (g/dL) | 4.2 ± 0.4 | 4.1 ± 0.4 | 3.9 ± 0.5 | <0.001 |
| Uric acid (mg/dL) | 6.7 ± 2.7 | 7.6 ± 3.1 | 8.7 ± 2.9 | 0.004 |
| Calcium (mg/dL) | 9.1 ± 0.4 | 9.0 ± 0.4 | 8.9 ± 0.4 | <0.001 |
| Phosphate (mg/dL) | 3.4 ± 0.5 | 4.0 ± 0.8 | 4.7 ± 1.0 | <0.001 |
| Total cholesterol (mg/dL) | 208.8 ± 39.9 | 209.4 ± 41.4 | 207.6 ± 40.4 | 0.892 |
| Hemoglobin (g/dL) | 12.9 ± 1.7 | 10.9 ± 1.6 | 9.7 ± 1.5 | <0.001 |
| CRP (mg/dL) | 0.6 ± 0.5 | 0.7 ± 0.5 | 1.0 ± 0.9 | <0.001 |
| Intact PTH (pg/mL) | 56.5 ± 29.6 | 129.0 ± 74.0 | 196.2 ± 90.9 | <0.001 |
| LVMI (g/m$^2$) | 95.0 ± 23.0 | 105.0 ± 24.3 | 113.8 ± 23.8 | <0.001 |
| E (cm/s) | 62.0 ± 8.4 | 61.2 ± 8.3 | 61.0 ± 10.2 | 0.230 |
| e' (cm/s) | 7.9 ± 1.4 | 7.6 ± 1.1 | 7.3 ± 1.3 | <0.001 |
| E/e' | 8.1 ± 1.7 | 8.2 ± 1.3 | 8.6 ± 2.4 | <0.001 |
| Left atrial volume index (mL/m$^2$) | 34 ± 6.9 | 35.3 ± 6.6 | 36.1 ± 7.6 | 0.014 |
| LVEF (%) | 62.3 ± 6.7 | 61.3 ± 7.4 | 60.0 ± 7.6 | 0.001 |
| LVH | 153 (29.9%) | 167 (46.9%) | 105 (66.5%) | <0.001 |
| LVDD | 316 (61.8%) | 232 (65.2%) | 109 (69.0%) | 0.229 |

Data are presented as mean ± standard deviation or (n, %).

[a]Coronary heart disease is defined as a history of coronary artery bypass surgery or percutaneous transluminal coronary angioplasty.

[b]Cerebrovascular disease is defined as a history of stroke or transient ischemic attack.

[c]P indicates statistical significance between the 3 groups. ACEI, angiotensin-converting enzyme inhibitor; ARB, angiotensin receptor blocker; CKD, chronic kidney disease; CRP, C-reactive protein; E, early mitral inflow velocity; e', peak early mitral annular velocity; eGFR, estimated glomerular filtration rate; PTH, parathyroid hormone; LVDD, left ventricular diastolic dysfunction; LVEF, left ventricular ejection fraction; LVH, left ventricular hypertrophy; LVMI, left ventricular mass index.

ratio (P < 0.001) and lower values of e' (P < 0.001). They tended to have a higher prevalence of LVH (P < 0.001) and LVDD (P < 0.001). The baseline characteristics of the study population according to systolic blood pressure are shown in Table 3. Patients with a

**Table 2. Baseline characteristics of the study population according to tertiles of serum uric acid values (n = 1025).**

| | Tertile 1 | Tertile 2 | Tertile 3 | P[c] |
|---|---|---|---|---|
| | (<5.5 mg/dL) | (5.6–8.4 mg/dL) | (>8.4 mg/dL) | |
| Age (years) | 59.0 ± 10.0 | 59.8 ± 10.4 | 61.9 ± 10.8 | 0.001 |
| Sex, male [n (%)] | 170 (50.1%) | 196 (57.6%) | 173 (50.0%) | 0.073 |
| Diabetes [n (%)] | 182 (53.7%) | 171 (50.3%) | 169 (48.8%) | 0.430 |
| Cardiovascular disease [n (%)] | | | | |
| Coronary heart disease[a] | 61 (18.0%) | 75 (22.1%) | 78 (22.5%) | 0.276 |
| Cerebrovascular disease[b] | 31 (9.1%) | 37 (10.9%) | 44 (12.7%) | 0.325 |
| Peripheral vascular disease | 22 (6.5%) | 24 (7.1%) | 34 (9.8%) | 0.218 |
| Medication [n (%)] | | | | |
| ACEI or ARB | 250 (73.7%) | 265 (77.9%) | 263 (76.0%) | 0.441 |
| Calcium channel blocker | 208 (61.4%) | 213 (62.6%) | 223 (64.5%) | 0.701 |
| Beta blocker | 125 (36.9%) | 131 (38.5%) | 145 (41.9%) | 0.387 |
| Diuretics (thiazide) | 108 (31.9%) | 98 (28.8%) | 81 (23.4%) | 0.137 |
| Diuretics (loop) | 143 (42.2%) | 145 (42.6%) | 166 (48.0%) | 0.236 |
| Urate-lowering therapy | 78 (23.0%) | 47 (13.8%) | 73 (21.1%) | 0.006 |
| Gout | 42 (12.4%) | 49 (14.4%) | 71 (20.5%) | 0.010 |
| Body mass index (kg/m$^2$) | 23.4 ± 2.5 | 23.7 ± 2.5 | 23.5 ± 2.6 | 0.280 |
| Systolic blood pressure (mmHg) | 130.2 ± 17.6 | 132.6 ± 18.2 | 137.7 ± 19.3 | <0.001 |
| Diastolic blood pressure (mmHg) | 80.2 ± 14.2 | 80.0± 14.4 | 80.2 ± 15.2 | 0.982 |
| eGFR (mL/min/1.73 m$^2$) | 32.7 ± 13.1 | 32.2 ± 15.0 | 26.9 ± 14.4 | <0.001 |
| Urinary albumin (mg/g Cr) | 1030.1 ± 1019.3 | 1270.9 ± 1196.0 | 1359.1 ± 1250.9 | 0.001 |
| Albumin (g/dL) | 4.2 ± 0.4 | 4.1 ± 0.4 | 4.0 ± 0.5 | 0.007 |
| Calcium (mg/dL) | 9.1 ± 0.4 | 9.1 ± 0.4 | 9.0 ± 0.4 | 0.231 |
| Phosphate (mg/dL) | 3.6 ± 0.8 | 3.6 ± 0.8 | 4.0 ± 0.9 | <0.001 |
| Total cholesterol (mg/dL) | 209.8 ± 39.9 | 209.8 ± 42.5 | 206.9 ± 39.1 | 0.556 |
| Hemoglobin (g/dL) | 12.0 ± 2.0 | 11.7 ± 2.1 | 11.3 ± 2.0 | <0.001 |
| CRP (mg/dL) | 0.6 ± 0.5 | 0.7 ± 0.6 | 0.8 ± 0.6 | 0.031 |
| Intact PTH (pg/mL) | 82.7 ± 63.8 | 99.1 ± 78.5 | 127.4 ± 86.6 | <0.001 |
| LVMI (g/m$^2$) | 87.5 ± 18.3 | 101.4 ± 24.2 | 115.0 ± 22.7 | <0.001 |
| E (cm/s) | 63.8 ± 8.5 | 61.1 ± 7.7 | 60.0 ± 9.3 | <0.001 |
| e' (cm/s) | 8.6 ± 1.3 | 7.6 ± 1.2 | 6.9 ± 1.0 | <0.001 |
| E/e' | 7.6 ± 1.4 | 8.3 ± 1.6 | 8.8 ± 1.8 | <0.001 |
| Left atrial volume index (mL/m$^2$) | 30.0 ± 6.4 | 35.7 ± 6.7 | 38.9 ± 4.5 | <0.001 |
| LVEF (%) | 61.6 ± 7.1 | 61.6 ± 6.9 | 61.7 ± 7.4 | 0.950 |
| LVH | 41 (12.1%) | 139 (40.9%) | 245 (70.8%) | <0.001 |
| LVDD | 91 (26.8%) | 239 (70.3%) | 327 (94.5%) | <0.001 |

Data are presented as mean ± standard deviation or (n, %).

[a]Coronary heart disease is defined as a history of coronary artery bypass surgery or percutaneous transluminal coronary angioplasty.

[b]Cerebrovascular disease is defined as a history of stroke or transient ischemic attack.

[c]P indicates statistical significance between the 3 groups. ACEI, angiotensin-converting enzyme inhibitor; ARB, angiotensin receptor blocker; CKD, chronic kidney disease; CRP, C-reactive protein; E, early mitral inflow velocity; e', peak early mitral annular velocity, eGFR, estimated glomerular filtration rate; PTH, parathyroid hormone; LVDD, left ventricular diastolic dysfunction; LVEF, left ventricular ejection fraction; LVH, left ventricular hypertrophy; LVMI, left ventricular mass index.

higher tertile of systolic blood pressure were more likely to have higher levels of phosphate (P < 0.001), intact PTH (P < 0.001), LVMI (P < 0.001), E/e' (P < 0.001), and left atrial volume index (P < 0.001) and lower levels of eGFR (P < 0.001), hemoglobin (P < 0.001), e' (P < 0.001), and LVEF (P = 0.041). They were also likely to have a higher prevalence of

**Table 3. Baseline characteristics of the study population according to tertiles of systolic blood pressure values (n = 1025).**

| | Tertile 1 (<129 mmHg) | Tertile 2 (129–141 mmHg) | Tertile 3 (>141 mmHg) | P[c] |
|---|---|---|---|---|
| Age (years) | 59.3 ± 10.7 | 60.9 ± 10.5 | 60.6 ± 10.2 | 0.109 |
| Sex, male [n (%)] | 180 (53.4%) | 170 (48.0%) | 172 (51.5%) | 0.355 |
| Diabetes [n (%)] | 182 (53.7%) | 171 (50.3%) | 169 (48.8%) | 0.430 |
| Cardiovascular disease [n (%)] | | | | |
| Coronary heart disease[a] | 73 (21.7%) | 61 (17.2%) | 80 (24.0%) | 0.087 |
| Cerebrovascular disease[b] | 34 (10.1%) | 33 (9.3%) | 45 (13.5%) | 0.182 |
| Peripheral vascular disease | 23 (6.8%) | 25 (6.9%) | 33 (9.9%) | 0.227 |
| Medication [n (%)] | | | | |
| ACEI or ARB | 257 (76.3%) | 271 (76.6%) | 250 (74.9%) | 0.857 |
| Calcium channel blocker | 202 (59.9%) | 222 (62.7%) | 223 (63.1%) | 0.283 |
| Beta blocker | 132 (39.2%) | 131 (38.9%) | 145 (41.3%) | 0.511 |
| Diuretics (thiazide) | 113 (33.5%) | 92 (26.0%) | 82 (24.6%) | 0.020 |
| Diuretics (loop) | 156 (46.3%) | 143 (40.4%) | 155 (46.4%) | 0.189 |
| Urate-lowering therapy | 61 (18.1%) | 73 (20.6%) | 64 (19.2%) | 0.701 |
| Gout | 50 (14.8%) | 54 (15.3%) | 58 (17.4%) | 0.628 |
| Body mass index (kg/m$^2$) | 23.7 ± 2.5 | 23.5 ± 2.5 | 23.4 ± 2.3 | 0.293 |
| eGFR (mL/min/1.73 m$^2$) | 35.6 ± 13.9 | 30.4 ± 13.6 | 25.7 ± 14.0 | <0.001 |
| Urinary albumin (mg/g Cr) | 1038.8 ± 1076.8 | 1321.2 ± 1246.8 | 1298.6 ± 1150.4 | 0.002 |
| Albumin (g/dL) | 4.2 ± 0.4 | 4.1 ± 0.5 | 4.1 ± 0.4 | 0.009 |
| Calcium (mg/dL) | 9.1 ± 0.4 | 9.0 ± 0.4 | 9.0 ± 0.4 | 0.306 |
| Phosphate (mg/dL) | 3.6 ± 0.7 | 3.8 ± 0.9 | 3.9 ± 0.9 | <0.001 |
| Total cholesterol (mg/dL) | 209.2 ± 38.8 | 207.8 ± 41.1 | 209.6 ± 41.5 | 0.832 |
| Hemoglobin (g/dL) | 12.4 ± 1.9 | 11.5 ± 2.2 | 11.2 ± 1.9 | <0.001 |
| CRP (mg/dL) | 0.6 ± 0.4 | 0.7 ± 0.6 | 0.8 ± 0.7 | 0.019 |
| Intact PTH (pg/mL) | 78.7 ± 58.3 | 103.8 ± 77.7 | 127.4 ± 90.6 | <0.001 |
| LVMI (g/m$^2$) | 95.2 ± 23.7 | 98.8 ± 23.5 | 110.4 ± 24.0 | <0.001 |
| E (cm/s) | 61.7 ± 8.3 | 61.3 ± 8.8 | 61.8 ± 9.0 | 0.722 |
| e' (cm/s) | 7.9 ± 1.3 | 7.7 ± 1.3 | 7.4 ± 1.3 | <0.001 |
| E/e' | 7.9 ± 1.5 | 8.1 ± 1.6 | 8.6 ± 1.8 | <0.001 |
| Left atrial volume index (mL/m$^2$) | 33.7 ± 7.0 | 34.9 ± 6.9 | 36.5 ± 6.7 | <0.001 |
| LVEF (%) | 62.4 ± 7.1 | 61.3 ± 7.1 | 61.2 ± 7.1 | 0.041 |
| LVH | 100 (29.7%) | 128 (36.2%) | 197 (59.0%) | <0.001 |
| LVDD | 186 (55.2%) | 229 (64.7%) | 242 (72.5%) | <0.001 |

Data are presented as mean ± standard deviation or (n, %).

[a]Coronary heart disease is defined as a history of coronary artery bypass surgery or percutaneous transluminal coronary angioplasty.

[b]Cerebrovascular disease is defined as a history of stroke or transient ischemic attack.

[c]P indicates statistical significance between the 3 groups. ACEI, angiotensin-converting enzyme inhibitor; ARB, angiotensin receptor blocker; CKD, chronic kidney disease; CRP, C-reactive protein; E, early mitral inflow velocity; e', peak early mitral annular velocity, eGFR, estimated glomerular filtration rate; PTH, parathyroid hormone; LVDD, left ventricular diastolic dysfunction; LVEF, left ventricular ejection fraction; LVH, left ventricular hypertrophy; LVMI, left ventricular mass index.

LVH (P < 0.001) and LVDD (P < 0.001). The association of SUA levels with the LVMI, left atrial volume index, e', and E/e' ratio is shown in Fig 1. SUA levels correlated positively with the LVMI (r = 0.483, P < 0.001), E/e' ratio (r = 0.302, P < 0.001), and left atrial volume index (r = 0.521, P < 0.001) and negatively with e' (r = -0.520, P < 0.001).

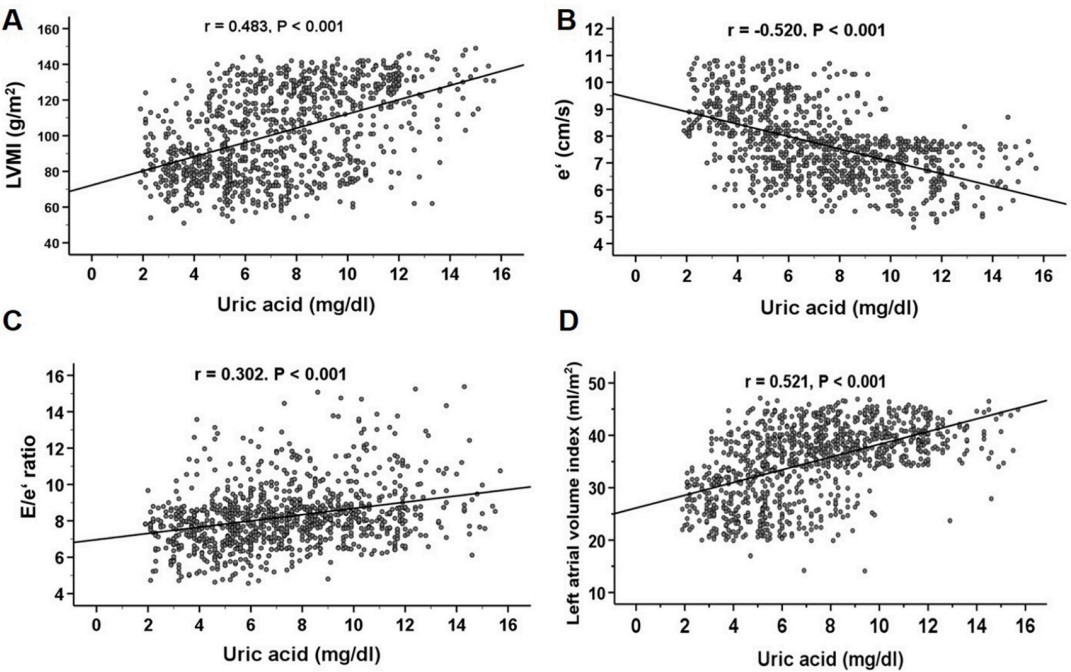

**Fig 1.** Correlations between SUA levels and LVMI (**A**), e' (**B**), E/e' (**C**), and left atrial volume index (**D**) in patients with pre-dialysis CKD (n = 1025). The SUA level correlated positively with the LVMI, E/e' ratio, and left atrial volume index and correlated negatively with the e'. CKD, chronic kidney disease; E, early mitral inflow velocity; e', peak early mitral annular velocity; LVMI, left ventricular mass index; SUA, serum uric acid.

## Association between SUA levels and LVH

Table 4 shows the baseline variables that were found to be associated with the presence of LVH in the study subjects. In the univariable analysis, the predictors of LVH were as follows: age (OR: 1.04, 95% CI: 1.03–1.05, P < 0.001), coronary heart disease (OR: 1.55, 95% CI: 1.15–2.10, P = 0.005), gout (OR: 1.47, 95% CI: 1.05–2.05, P = 0.026), systolic blood pressure (OR: 1.33, 95% CI: 1.23–1.43, P < 0.001), diastolic blood pressure (OR: 1.11, 95% CI: 1.02–1.21, P = 0.016), eGFR (OR: 0.97, 95% CI: 0.96–0.98, P < 0.001), urinary albumin levels (OR: 1.02, 95% CI: 1.01–1.03, P = 0.002), serum albumin levels (OR: 0.71, 95% CI: 0.53–0.95, P = 0.023), SUA levels (OR: 1.56, 95% CI: 1.47–1.65, P < 0.001), phosphate (OR: 1.47, 95% CI: 1.27–1.71, P< 0.001), hemoglobin (OR: 0.85, 95% CI: 0.80–0.91, P < 0.001), CRP (OR: 1.44, 95% CI: 1.15–1.80, P = 0.001), intact PTH (OR: 1.09, 95% CI: 1.07–1.11, P < 0.001), and left atrial volume index (OR: 1.16, 95% CI: 1.13–1.18, P < 0.001). In the multivariable analysis, the SUA level (OR: 1.40, 95% CI: 1.31–1.50, P < 0.001) was an independent predictor of LVH. In addition, age (OR: 1.03, 95% CI: 1.01–1.05, P < 0.001), systolic blood pressure (OR: 1.24, 95% CI: 1.11–1.38, P < 0.001), and levels of intact PTH (OR: 1.06, 95% CI: 1.03–1.09, P < 0.001) and left atrial volume index (OR: 1.80, 95% CI: 1.50–1.11, P < 0.001) were independent predictors of LVH.

## Association between SUA levels and LVDD

Table 5 shows the associations of the presence of LVDD with baseline variables in the study subjects. In the univariable analysis, the predictors of LVDD were as follows: age (OR: 1.02, 95% CI: 1.00–1.03, P = 0.012), systolic blood pressure (OR: 1.22, 95% CI: 1.13–1.30, P < 0.001), diastolic blood pressure (OR: 1.14, 95% CI: 1.05–1.25, P = 0.003), eGFR (OR: 0.99,

**Table 4. Univariable and multivariable analyses for variables associated with LVH in study population (n = 1025).**

| | Univariable | | Multivariable | |
|---|---|---|---|---|
| | Odds ratio (95% CI) | P | Odds ratio (95% CI) | P |
| Age (1 year) | 1.04 (1.03–1.05) | <0.001 | 1.03 (1.01–1.05) | <0.001 |
| Sex, male | 0.95 (0.74–1.21) | 0.658 | 0.99 (1.00–0.73) | 0.991 |
| Diabetes | 0.86 (0.67–1.10) | 0.231 | 0.91 (0.67–1.25) | 0.557 |
| Cardiovascular disease | | | | |
| Coronary heart disease[a] | 1.55 (1.15–2.10) | 0.005 | 1.52 (1.04–2.25) | 0.033 |
| Cerebrovascular disease[b] | 1.42 (0.96–2.10) | 0.083 | | |
| Peripheral vascular disease | 1.24 (0.78–1.95) | 0.366 | | |
| Medication | | | | |
| ACEI or ARB | 0.92 (0.69–1.24) | 0.595 | | |
| Calcium channel blocker | 1.15 (0.89–1.49) | 0.296 | | |
| Beta blocker | 1.08 (0.84–1.40) | 0.539 | | |
| Diuretics (thiazide) | 0.79 (0.59–1.04) | 0.091 | | |
| Diuretics (loop) | 1.15 (0.90–1.48) | 0.264 | | |
| Urate-lowering therapy | 1.26 (0.92–1.72) | 0.153 | | |
| Gout | 1.47 (1.05–2.05) | 0.026 | 1.08 (0.70–1.67) | 0.723 |
| Body mass index (1 kg/m$^2$) | 0.97 (0.93–1.03) | 0.315 | | |
| Systolic blood pressure (10 mmHg) | 1.33 (1.23–1.43) | <0.001 | 1.24 (1.11–1.38) | <0.001 |
| Diastolic blood pressure (10 mmHg) | 1.11 (1.02–1.21) | 0.016 | 0.89 (0.78–1.01) | 0.068 |
| eGFR (1 ml/min/1.73 m$^2$) | 0.97 (0.96–0.98) | <0.001 | 1.00 (0.99–1.02) | 0.663 |
| Urinary albumin (100 mg/g Cr) | 1.02 (1.01–1.03) | 0.002 | 1.01 (0.97–1.06) | 0.622 |
| Albumin (1 g/dL) | 0.71 (0.53–0.95) | 0.023 | 1.33 (0.43–4.14) | 0.622 |
| Uric acid (1 mg/dL) | 1.56 (1.47–1.65) | <0.001 | 1.40 (1.31–1.50) | <0.001 |
| Calcium (1 mg/dL) | 0.88 (0.64–1.21) | 0.427 | | |
| Phosphate (1 mg/dL) | 1.47 (1.27–1.71) | <0.001 | 0.93 (0.75–1.15) | 0.495 |
| Total cholesterol (1 mg/dL) | 1.00 (1.00–1.00) | 0.205 | | |
| Hemoglobin (1 g/dL) | 0.85 (0.80–0.91) | <0.001 | 1.00 (0.91–1.10) | 1.000 |
| CRP (1 mg/dL) | 1.44 (1.15–1.80) | 0.001 | 1.08 (0.83–1.42) | 0.568 |
| Intact PTH (10 pg/mL) | 1.09 (1.07–1.11) | <0.001 | 1.06 (1.03–1.09) | <0.001 |
| Left atrial volume index (1 mL/m$^2$) | 1.16 (1.13–1.18) | <0.001 | 1.08 (1.50–1.11) | <0.001 |

[a]Coronary heart disease is defined as a history of coronary artery bypass surgery or percutaneous transluminal coronary angioplasty.

[b]Cerebrovascular disease is defined as a history of stroke or transient ischemic attack. ACEI, angiotensin-converting enzyme inhibitor; ARB, angiotensin receptor blocker; CI, confidence interval; CRP, C-reactive protein; eGFR, estimated glomerular filtration rate; PTH, parathyroid hormone; LVH, left ventricular hypertrophy.

95% CI: 0.98–1.00, P 0.010), SUA levels (OR: 1.98, 95% CI: 1.82–2.16, P < 0.001), phosphate levels (OR: 1.26, 95% CI: 1.08–1.48, P 0.003), hemoglobin levels (OR: 0.94, 95% CI: 0.88–1.00, P = 0.047), and levels of intact PTH and (OR: 1.05, 95% CI: 1.03–1.07, P < 0.001) and left atrial volume index (OR: 3.74, 95% CI: 2.90–4.81, P < 0.001). In the multivariable analysis, the SUA level (OR: 1.93, 95% CI: 1.53–2.43, P < 0.001) was an independent predictor of LVDD. Furthermore, systolic blood pressure (OR: 1.42, 95% CI: 1.05–1.92, P = 0.023) and left atrial volume index (OR: 3.09, 95% CI: 2.45–3.90, P < 0.001) were an independent predictor of LVDD.

## Investigation of the diagnostic power of SUA levels for predicting LVH and LVDD

ROC analysis was performed to investigate the diagnostic power of SUA levels for predicting the presence of LVH and LVDD in the study subjects (Fig 2). The AUC for SUA levels was

**Table 5. Univariable and multivariable analyses for variables associated with LVDD in study population (n = 1025).**

| | Univariable | | Multivariable | |
|---|---|---|---|---|
| | Odds ratio (95% CI) | P | Odds ratio (95% CI) | P |
| Age (1 year) | 1.02 (1.00–1.03) | 0.012 | 0.99 (0.95–1.03) | 0.619 |
| Sex, male | 0.99 (0.77–1.28) | 0.949 | 1.48 (0.63–3.48) | 0.365 |
| Diabetes | 1.08 (0.84–1.39) | 0.566 | 1.62 (0.67–3.92) | 0.288 |
| Cardiovascular disease | | | | |
| Coronary heart disease[a] | 1.13 (0.83–1.56) | 0.439 | 1.74 (0.57–5.25) | 0.328 |
| Cerebrovascular disease[b] | 1.10 (0.73–1.67) | 0.645 | | |
| Peripheral vascular disease | 1.18 (0.72–1.92) | 0.509 | | |
| Medication | | | | |
| ACEI or ARB | 0.98 (0.73–1.33) | 0.918 | | |
| Calcium channel blocker | 1.10 (0.85–1.43) | 0.483 | | |
| Beta blocker | 0.97 (0.74–1.25) | 0.786 | | |
| Diuretics (thiazide) | 0.90 (0.68–1.20) | 0.472 | | |
| Diuretics (loop) | 1.00 (0.77–1.29) | 1.000 | | |
| Urate-lowering therapy | 0.88 (0.64–1.21) | 0.418 | | |
| Gout | 1.18 (0.83–1.69) | 0.357 | | |
| Body mass index (1 kg/m$^2$) | 0.98 (0.93–1.04) | 0.498 | | |
| Systolic blood pressure (10 mmHg) | 1.22 (1.13–1.30) | <0.001 | 1.42 (1.05–1.92) | 0.023 |
| Diastolic blood pressure (10 mmHg) | 1.14 (1.05–1.25) | 0.003 | 0.69 (0.47–0.99) | 0.055 |
| eGFR (1 mL/min/1.73 m$^2$) | 0.99 (0.98–1.00) | 0.010 | 1.01 (0.97–1.06) | 0.554 |
| Urinary albumin (100 mg/g Cr) | 1.01 (1.00–1.02) | 0.070 | | |
| Albumin (1 g/dL) | 0.83 (0.60–1.12) | 0.221 | | |
| Uric acid (1 mg/dL) | 1.98 (1.82–2.16) | <0.001 | 1.93 (1.53–2.43) | <0.001 |
| Calcium (1 mg/dL) | 0.90 (0.65–1.24) | 0.513 | | |
| Phosphate (1 mg/dL) | 1.26 (1.08–1.48) | 0.003 | 0.64 (0.35–1.17) | 0.144 |
| Total cholesterol (1 mg/dL) | 1.00 (1.00–1.00) | 0.662 | | |
| Hemoglobin (1 g/dL) | 0.94 (0.88–1.00) | 0.047 | 0.97 (0.74–1.27) | 0.798 |
| CRP (1 mg/dL) | 1.15 (0.92–1.45) | 0.230 | | |
| Intact PTH (10 pg/mL) | 1.05 (1.03–1.07) | <0.001 | 1.01 (0.94–1.09) | 0.711 |
| Left atrial volume index (1 mL/m$^2$) | 3.74 (2.90–4.81) | <0.001 | 3.09 (2.45–3.90) | <0.001 |

[a]Coronary heart disease is defined as a history of coronary artery bypass surgery or percutaneous transluminal coronary angioplasty.

[b]Cerebrovascular disease is defined as a history of stroke or transient ischemic attack. ACEI, angiotensin-converting enzyme inhibitor; ARB, angiotensin receptor blocker; CI, confidence interval; CRP, C-reactive protein; eGFR, estimated glomerular filtration rate; PTH, parathyroid hormone; LVDD, left ventricular diastolic dysfunction.

0.803 (95% CI: 0.777–0.827) for LVH. The best cutoff value of SUA levels for predicting the presence of LVH was ≥ 7.5 mg/dL with the associated sensitivity of 71.8% (95% CI: 67.2–76.0%) and specificity of 75.2% (95% CI: 71.5–78.6%). Notably, combination of SUA and other significant factors in multivariable analysis (SBP and intact PTH) exhibited AUC of 0.836 (95% CI: 0.812–0.860), which was significantly higher than that of SUA alone (0.836 vs. 0.803, P < 0.001).

The AUC for SUA levels was 0.867 (95% CI: 0.845–0.887) for LVDD. The best cutoff value of SUA levels for predicting the presence of LVDD was ≥ 6.3 mg/dL with an associated sensitivity of 78.4% (95% CI: 75.0–81.5%) and specificity of 79.4% (95% CI: 74.8–83.4%). However, the combination of SUA and other significant factor in multivariable analysis (SBP) had a comparable AUC (0.871, 95% CI: 0.849–0.892) with SUA alone (0.871 vs. 0.867, P = 0.136).

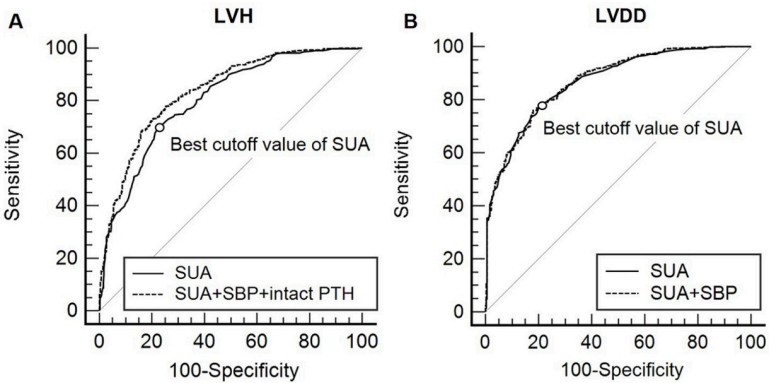

**Fig 2.** Receiver-operating characteristic curves of SUA levels for predicting the presence of LVH **(A)** and LVDD **(B)** in patients with pre-dialysis CKD (n = 1025). **(A)** The AUC for SUA levels was 0.803 (95% CI: 0.777–0.827) for LVH. The best cutoff value of SUA levels for predicting the presence of LVH was ≥ 7.5 mg/dL with the associated sensitivity of 71.8% (95% CI: 67.2–76.0%) and specificity of 75.2% (95% CI: 71.5–78.6%). Notably, combination of SUA, SBP, and intact PTH exhibited AUC of 0.836 (95% CI: 0.812–0.860), which was significantly higher than that of SUA alone (0.836 vs. 0.803, P < 0.001). **(B)** The AUC for SUA levels was 0.867 (95% CI: 0.845–0.887) for LVDD. The best cutoff value of SUA levels for predicting the presence of LVDD was ≥ 6.3 mg/dL with an associated sensitivity of 78.4% (95% CI: 75.0–81.5%) and specificity of 79.4% (95% CI: 74.8–83.4%). However, the combination of SUA and SBP had a comparable AUC (0.871, 95% CI: 0.849–0.892) with SUA alone (0.871 vs. 0.867, P = 0.136). AUC, area under the curve; CI, confidence interval; CKD, chronic kidney disease; LVDD, left ventricular diastolic dysfunction; LVH, left ventricular hypertrophy; PTH, parathyroid hormone; SBP, systolic blood pressure; SUA, serum uric acid.

## Discussion

LVH and LVDD frequently occur in patients with CKD and are known to be independent risk factors for future cardiovascular morbidity and mortality in these patients [2, 4, 16]. The prevalence of LVH increases with declining renal function in patients with CKD [16]. However, there have been no reports about the prevalence of LVDD with regard to the renal function of patients. In the present study, we found that the prevalence of LVH increased as the CKD stage increased. The prevalence of LVDD tended to increase with increasing CKD stages. However, the result was not statistically different between the three CKD stages. The severity of LVH and LVDD also increased with advanced CKD stages. Patients with higher CKD stages showed a higher LVMI and E/e' ratio and a lower e' value.

Uric acid is primarily associated with gout. However, during the past decades, uric acid itself has been reported to be a risk factor for CVDs in various populations, including the general population with no comorbidities and those with hypertension, congestive heart failure, and diabetes [17]. Of all CVDs, SUA levels have been reported to be associated with structural and functional cardiac diseases, including LVH and LVDD. Marotta et al. showed that among 557 healthy subjects, men with higher SUA levels (≥ 5.5 mg/dL) showed higher LV mass than men with lower SUA levels [18]. Fujita et al. reported that SUA levels were independently associated with LVH with an OR of 2.79 in 116 male patients with cardiac diseases [9]. Yamauchi et al. showed that SUA levels were associated with LVH, independent of confounding factors, including fibroblast growth factor (FGF) 23 and diuretics in 219 and 519 female and male patients with cardiac diseases, respectively, who were free from uric acid-lowering medications [19]. Concerning LVDD, Cicoira et al. reported that increased SUA levels were associated with LVDD in 150 patients with dilated cardiomyopathy [20]. Nogi et al. showed that among 744 patients having cardiac diseases with preserved ejection fraction, SUA levels were significantly associated with LVDD in women but not in men [11]. In another study, Lin et al. reported that gout, but not hyperuricemia, is associated with LVDD in 173 patients [21]. However, despite

the high prevalence of both, hyperuricemia and LVH/LVDD, in patients with CKD, studies investigating the association between SUA levels and LVH/LVDD have been scarce. Thus, we investigated this association in patients with CKD.

The main finding of the present study is that an elevated SUA level is an independent predictor of LVH and LVDD in patients with CKD. However, the present study is not the first to report the association between SUA levels and LVH/LVDD in the CKD population. Zeng et al. reported that elevated SUA levels were positively associated with an increased risk of LVH in CKD patients with type 2 diabetes [22]. Yoshitomi et al. reported that SUA levels were associated with LVMI and LVH in female patients with CKD, whereas no such association was found in male patients with CKD [16]. In contrast to the two above-mentioned studies [16, 22], the multivariable analysis in our study showed an association between SUA levels and LVH in patients with CKD, independent of diabetes or sex, suggesting that the SUA level is an independent predictor of LVH in the CKD population.

Concerning LVDD in patients with CKD, Gromadziński et al. showed that hyperuricemia was an independent predictor of LVDD in 50 patients with CKD [10]. However, their study was limited by the small sample size. Our study showed an independent association between SUA levels and LVDD in 1025 patients with CKD, a relatively large sample size meeting the statistical significance.

The mechanism underlying LVH and LVDD in patients with CKD is unclear. LVH and LVDD are closely related, and the primary mechanism of LVDD is LVH with myocardial fibrosis, which induces myocardial stiffness and impairs cardiac function during diastole [23]. LVH in CKD is a physiological response to pressure and volume overload [23]. Sustained pressure/volume overload and uremia-related factors such as anemia, hyperparathyroidism, chronic inflammation, and levels of FGF 23 have been suggested to play a role in the mechanism underlying the development of LVH in patients with CKD [23]. In this study, consistent with the mechanisms suggested above, we found that systolic blood pressure and levels of intact PTH are independent risk factors for LVH.

The mechanism of an independent association between SUA levels and LVH/LVDD is unclear. However, previous studies have suggested a mechanism for these associations. First, it seems likely that one mechanism is the effect of elevated SUA levels on blood pressure [24]. However, the present study showed that the SUA level is independently associated with LVH and LVDD after adjustment for systolic/diastolic blood pressure. Second, previous experimental studies have suggested the direct role of SUA levels in LVH and LVDD [24]. Chen et al. reported that hyperuricemia is associated with increased myocardial oxidative stress, which contributes to ventricular remodeling and LVH. These changes were prevented by allopurinol, a xanthine oxidase inhibitor [25]. Engberding et al. showed that the expression of xanthine oxidase, a major source of reactive oxygen species, increased in the remote myocardium after myocardial infarction in mice. In that study, allopurinol treatment attenuated LV remodeling processes and dysfunction [26]. Jia et al. reported that in mice that were fed with a western diet, uric acid promoted LVH and LVDD via activation of the S6 kinase-1 growth pathway and profibrotic transforming growth factor-β1, along with macrophage proinflammatory polarization. Allopurinol treatment prevented these adverse changes [27].

The present study has several limitations. First, owing to its retrospective and cross-sectional design, it is difficult to establish the temporal relationship and causality between SUA levels and LVH/LVDD. We believe that future prospective clinical and experimental studies are needed to establish the causal relationship between SUA levels and LVH/LVDD in patients with CKD. Second, there was a selection bias in the inclusion of study subjects in this study. We only included CKD patients with preserved LV systolic function and excluded those with valvular heart disease, congenital heart disease, cardiomyopathy, and atrial fibrillation to reveal

the association between SUV levels and LVH/LVDD more clearly. Therefore, the results of our study may not be extrapolated to the overall CKD population. Third, the present study was a cross-sectional study investigating the association between SUA levels and echocardiographic findings. Thus, the timing of the measurement of SUA levels and echocardiography is essential. However, not all measurements of SUA levels were performed on the same day of echocardiography. The mean interval from the SUA level measurement to echocardiography was 5.6 ± 3.1 days (range: 0.1–11.3 days).

Despite these limitations, our study has important clinical implications compared to the previous studies which demonstrated the association between SUA and LVH/LVDD in the CKD population. First, cardiorenal syndrome (CRS) has gained considerable attention. CRS encompasses conditions in which failure of either the heart or the kidney leads to or accelerates other organ failures [28]. Type-4 CRS, also defined as a chronic reno-cardiac disease, is characterized by primary CKD leading to an impairment of cardiac function, LVH, LVDD, or increased risk of adverse cardiovascular events [29]. In the Framingham Heart Study cohort, the SUA level has been reported to be a marker for subsequent LV systolic function in 2269 participants without congestive heart failure [30]. In the present study, LVH and LVDD in patients with CKD were predicted to be associated with SUA levels of $\geq$ 7.5 mg/dL and $\geq$ 6.3 mg/dL, respectively. Thus, based on the SUA level observed in LV systolic dysfunction, we believe that SUA could be a biomarker for LVH and LVDD in type-4 CRS. Second, there has been a lot of research that demonstrated the adverse effect of SUA on LVH/LVDD. However, as discussed above, there have been only a few studies on the association between SUA and LVH/LVDD in the CKD population. Although our study is not the first study to investigate these associations in the CKD population, our study has the strengths over the previous studies in that it included a large number of patients with CKD (n = 1025) and assessed a variety of variables that could affect the LVH/LVDD in patients with CKD, such as intact PTH, phosphate, left atrial volume, etc. Thus, our study provides more solid evidence for the association between SUA and LVH/LVDD in the CKD population and raises awareness of the importance of SUA during the development of LVH/LVDD in the CKD population.

In conclusion, we found that the SUA level is an independent predictor of LVH and LVDD in patients with CKD. Thus, we also showed the best cutoff value of SUA levels for predicting the presence of LVH and LVDD, suggesting that SUA could be a biomarker for LVH and LVDD in patients with CKD. Further clinical and experimental studies are needed to reveal the mechanism underlying this association and to determine whether uric acid-lowering agents can prevent the development of LVH and LVDD in patients with CKD.

## Author Contributions

**Conceptualization:** Il Young Kim, Soo Bong Lee.

**Data curation:** Il Young Kim, Byung Min Ye, Min Jeong Kim, Seo Rin Kim, Harin Rhee.

**Formal analysis:** Il Young Kim, Sang Heon Song, Eun Young Seong, Soo Bong Lee.

**Investigation:** Byung Min Ye, Min Jeong Kim, Hyo Jin Kim, Harin Rhee.

**Methodology:** Il Young Kim, Seo Rin Kim, Soo Bong Lee.

**Supervision:** Dong Won Lee, Sang Heon Song, Eun Young Seong.

**Validation:** Seo Rin Kim, Dong Won Lee, Hyo Jin Kim, Sang Heon Song, Eun Young Seong.

**Writing – original draft:** Il Young Kim.

**Writing – review & editing:** Soo Bong Lee.

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
