## [Decision Letter · Decision Letter 0]

25 Mar 2021

PONE-D-21-04044

Association between serum uric acid and left ventricular hypertrophy/left ventricular diastolic dysfunction in patients with chronic kidney disease

PLOS ONE

Dear Dr. Lee,

Thank you for submitting your manuscript to PLOS ONE. After careful consideration, we feel that it has merit but does not fully meet PLOS ONE’s publication criteria as it currently stands. Therefore, we invite you to submit a revised version of the manuscript that addresses the points raised during the review process.

Please not that reviewer #1 raised a very important point. Thus, the definition of left-ventricular diastolic dysfunction is obviously crucial for this paper and should follow the current recommendations. Both reviewers indicated that previous research addressed the aim of this study already. This does not mean that it is not interesting but the authors need to put their work better into perspective and need to better address previous studies in relation to their own work. This should include proper discussion about the new aspects of their own work.

We look forward to receiving your revised manuscript.

Kind regards,

Hans-Peter Brunner-La Rocca, M.D.

Academic Editor

PLOS ONE

Journal Requirements:

Reviewers' comments:

Reviewer's Responses to Questions

**Comments to the Author**

1. Is the manuscript technically sound, and do the data support the conclusions?

Reviewer #1: Yes

Reviewer #2: Yes

2. Has the statistical analysis been performed appropriately and rigorously? 

Reviewer #1: Yes

Reviewer #2: Yes

3. Have the authors made all data underlying the findings in their manuscript fully available?

Reviewer #1: No

Reviewer #2: Yes

4. Is the manuscript presented in an intelligible fashion and written in standard English?

Reviewer #1: Yes

Reviewer #2: Yes

5. Review Comments to the Author

Reviewer #1: 

MAJOR CONCERN

Despite the authors have mentioned the analysis of the left atrial volume to classify the left ventricular diastolic function (Line 108), I was not able to find these results in the text. This is a cardinal data, recommended by the American Society of Echocardiography Guidelines for the Evaluation of Left Ventricular diastolic Function (Nagueh SF et al. J Am Soc Echocardiogr 2016;29:277-314). The authors need to show these results and to include them in the uni and multivariate analysis. There are studies showing that high levels of serum uric acid are associated with higher left atrial volumes (Sung,KT et al. Association of serum uric acid level and gout with cardiac structure, function and sex differences from large scale asymptomatic Asians. PLOSOne.2020; 15(7):e0236173; Pan KL et al. The effects of gout on left atrial volume remodelling: a prospective echocardiographic study. Rheumatology.2014 May;53(5):867-74). The left atrial volume is a predictor of cardiovascular events (Mancusi C et al. Left atrial dilatation: A target organ damage in young to middle-age hypertensive patients. The Campania Salute Network. Int J Cardiol. 2018;265:299-233) and it would be important to investigate and show the correlation between this parameter and serum uric acid levels in this group of patients.

MINOR CONCERNS

Material and Methods

There is a need to define, even if briefly, the stages of Chronic Kidney Disease (CKD).

Line 105:

According to the American Society of Echocardiography Recommendations for the Evaluation of Left ventricular Diastolic Function Guidelines (Nagueh SF et al. J Am Soc Echocardiogr 2016;29:277-314) the correct is e´, instead of E´.

Lines 106-108:

“The E/E' ratio was calculated and used for the estimation of LV filling pressure. The severity of diastolic dysfunction was assessed using the E' values and E/E' ratios [13]. According to guideline of the American Society of Echocardiography [14]”.

It seems more appropriate to rewritten as follows:

“The E/e' ratio was calculated and used for the estimation of LV filling pressure. The severity of diastolic dysfunction was assessed using the e' values and E/e' ratio, [13] according to the American Society of Echocardiography Guidelines for the Evaluation of Left Ventricular diastolic Function. [14]”

Lines 108-110:

”According to guideline of the American Society of Echocardiography [14], E', left atrium volume, E/A, deceleration time of E, and E/E' was used to categorize diastolic dysfunction into normal function, or grades 1, 2, or 3 diastolic dysfunction.”

It seems more suitable to rewritten as follows:

“...was used to categorize diastolic function into normal or grades 1, 2,or 3....”

Lines 233-234:

“Notably, combination of SUA, SBP, and intact PTH) exhibited AUC of 0.836 (95% CI: 0.812–0.860), which was significant higher than that of SUA alone (0.836 vs. 0.803, P < 0.00”

There is no parenthesis after PTH.

...which was significantly higher than...

Reviewer #2: Dear Authors, this is a very nice work with more than 1000 patients. The association of uric acid with LVH or LVDD is unclear and for sure research has to be done. It is confused if LVH and diastolic dysfunction come only from the increased SBP and the renal failure, but the statistical analysis you provide gives strong support for the role of UA. I would like to ask why you didn't exclude CAD patients as well as it is established that many of these patients will also have diastolic dysfunction at baseline? Also, did you try to match your subgroups according to SBP which showed significant correlation with LVH and LVDD in multi-variable analysis? Finally, in tables 1 & 2 the p values refer to the overall comparison between the subgroups? The use of M-Mode for LV mass assessment seems a bit outdated but we can accept it as it in the ASE guidelines.

Language is proper

In pubmed there are a lot of publications for the role of SUA in LVH and LVDD. Is the content of your manuscript so strong to add valuable data in the literature? It is important that you have included so many patients and even if there are a lot similiar publications yours may add experience in this field and raise our awareness for research.

Thank you for your submission.

6. PLOS authors have the option to publish the peer review history of their article (what does this mean?). If published, this will include your full peer review and any attached files.

Reviewer #1: No

Reviewer #2: **Yes: **konstantinos Papadopoulos

---

## [Author Response · Author response to Decision Letter 0]

7 Apr 2021

PONE-D-21-04044

Association between serum uric acid and left ventricular hypertrophy/left ventricular diastolic dysfunction in patients with chronic kidney disease

Editor’s comment

Reviewer #1 raised a very important point. Thus, the definition of left-ventricular diastolic dysfunction is obviously crucial for this paper and should follow the current recommendations. Both reviewers indicated that previous research addressed the aim of this study already. This does not mean that it is not interesting but the authors need to put their work better into perspective and need to better address previous studies in relation to their own work. This should include proper discussion about the new aspects of their own work.

Answer)

First, we included the left atrium volume index in data analysis as reviewer #1’s comment

Second, as editor and reviewers comment, there has been a lot of research that demonstrated the adverse effect of SUA on LVH/LVDD. However, there have been only a few studies on the association between SUA and LVH/LVDD in the CKD population. Regarding the LVH in patients with CKD, Zeng et al. reported that elevated SUA levels were positively associated with an increased risk of LVH in CKD patients with type 2 diabetes [ J Diabetes Res. 2017; 2017: 5016093]. Yoshitomi et al. reported that SUA levels were associated with LVMI and LVH in female patients with CKD, whereas no such association was found in male patients with CKD [Hypertens Res. 2014; 37: 246-52]. In contrast to the two above-mentioned studies, the multivariable analysis in our study showed an association between SUA levels and LVH in patients with CKD, independent of diabetes or sex, suggesting that the SUA level is an independent predictor of LVH in the CKD population. Concerning LVDD in patients with CKD, Gromadzińsk et al. showed that hyperuricemia was an independent predictor of LVDD in 50 patients with CKD [Adv Clin Exp Med. 2015; 24: 47-54.]. However, their study was limited by the small sample size. Our study showed an independent association between SUA levels and LVDD in 1025 patients with CKD, a relatively large sample size meeting the statistical significance. Taken together, although our study is not the first study to investigate the association between SUA and LVH/LVDD in the CKD population, our study has the strengths over the previous studies in that it included a large number of patients with CKD (n = 1025) and assessed a variety of variables that could affect the LVH/LVDD in patients with CKD, such as intact PTH, phosphate, left atrial volume, etc. Thus, our study provides more solid evidence for the association between SUA and LVH/LVDD in the CKD population and raises awareness of the importance of SUA during the development of LVH/LVDD in the CKD population.

This description was added in the discussion section of the revised manuscript

Reviewer #1

MAJOR CONCERN

Despite the authors have mentioned the analysis of the left atrial volume to classify the left ventricular diastolic function (Line 108), I was not able to find these results in the text. This is a cardinal data, recommended by the American Society of Echocardiography Guidelines for the Evaluation of Left Ventricular diastolic Function (Nagueh SF et al. J Am Soc Echocardiogr 2016;29:277-314). The authors need to show these results and to include them in the uni and multivariate analysis. There are studies showing that high levels of serum uric acid are associated with higher left atrial volumes (Sung, KT et al. Association of serum uric acid level and gout with cardiac structure, function and sex differences from large scale asymptomatic Asians. PLOSOne.2020; 15(7):e0236173; Pan KL et al. The effects of gout on left atrial volume remodelling: a prospective echocardiographic study. Rheumatology.2014 May;53(5):867-74). The left atrial volume is a predictor of cardiovascular events (Mancusi C et al. Left atrial dilatation: A target organ damage in young to middle-age hypertensive patients. The Campania Salute Network. Int J Cardiol. 2018;265:299-233) and it would be important to investigate and show the correlation between this parameter and serum uric acid levels in this group of patients.

Answer) As you suggested, we included left atrial volume index in tables and figure. Please check the documents of response to reviewers

MINOR CONCERNS

Material and Methods

There is a need to define, even if briefly, the stages of Chronic Kidney Disease (CKD).

Answer) As you suggested, we added definition of the stages of CKD as follows.

Depending on the eGFR value, each patient was classified into one of the 3 CKD groups: CKD stage 3 (n = 511), 30 ≤ eGFR < 60 mL/min/1.73 m2; CKD stage 4 (n = 356), 15 ≤ eGFR <30 mL/min/1.73 m2; CKD stage 5 (n = 158), eGFR < 15 mL/min/1.73 m2.

Line 105:

According to the American Society of Echocardiography Recommendations for the Evaluation of Left ventricular Diastolic Function Guidelines (Nagueh SF et al. J Am Soc Echocardiogr 2016;29:277-314) the correct is e´, instead of E´.

Answer) We corrected E´ to e´.

Lines 106-108:

“The E/E' ratio was calculated and used for the estimation of LV filling pressure. The severity of diastolic dysfunction was assessed using the E' values and E/E' ratios [13]. According to guideline of the American Society of Echocardiography [14]”.

It seems more appropriate to rewritten as follows:

“The E/e' ratio was calculated and used for the estimation of LV filling pressure. The severity of diastolic dysfunction was assessed using the e' values and E/e' ratio, [13] according to the American Society of Echocardiography Guidelines for the Evaluation of Left Ventricular diastolic Function. [14]”

Answer) We revised as you suggested

Lines 108-110:

”According to guideline of the American Society of Echocardiography [14], E', left atrium volume, E/A, deceleration time of E, and E/E' was used to categorize diastolic dysfunction into normal function, or grades 1, 2, or 3 diastolic dysfunction.”

It seems more suitable to rewritten as follows:

“...was used to categorize diastolic function into normal or grades 1, 2,or 3....”

Answer) We revised as you suggested

Lines 233-234:

“Notably, combination of SUA, SBP, and intact PTH) exhibited AUC of 0.836 (95% CI: 0.812–0.860), which was significant higher than that of SUA alone (0.836 vs. 0.803, P < 0.00”

There is no parenthesis after PTH....Which was significantly higher than...

Answer) We revised as you suggested.

Reviewer #2

Dear Authors, this is a very nice work with more than 1000 patients. The association of uric acid with LVH or LVDD is unclear and for sure research has to be done. It is confused if LVH and diastolic dysfunction come only from the increased SBP and the renal failure, but the statistical analysis you provide gives strong support for the role of UA. I would like to ask why you didn't exclude CAD patients as well as it is established that many of these patients will also have diastolic dysfunction at baseline. Also, did you try to match your subgroups according to SBP which showed significant correlation with LVH and LVDD in multi-variable analysis? Finally, in tables 1 & 2 the p values refer to the overall comparison between the subgroups? The use of M-Mode for LV mass assessment seems a bit outdated but we can accept it as it in the ASE guidelines.

Language is proper

In PubMed, there are a lot of publications for the role of SUA in LVH and LVDD. Is the content of your manuscript so strong to add valuable data in the literature? It is important that you have included so many patients and even if there are a lot similar publications yours may add experience in this field and raise our awareness for research.

Thank you for your submission.

Comment 1) I would like to ask why you didn't exclude CAD patients as well as it is established that many of these patients will also have diastolic dysfunction at baseline.

Answer: We did not excluded CAD patients (n = 214) because loss of baseline data was severe if the CAD patients were excluded. Of the CAD patients (n = 214), 142 patients (66.4%) had the diastolic dysfunction at baseline. Thus, instead of excluding the CAD patients, we included the CAD patients in multivariable analysis for adjusting the effect of CAD on LVH and LVDD. 

Comment 2) Also, did you try to match your subgroups according to SBP which showed significant correlation with LVH and LVDD in multi-variable analysis

Answer) We add a table (Baseline characteristics of the study population according to tertiles of systolic blood pressure values). Please check the document for response to reviewer.

Comment 3) Finally, in tables 1 & 2 the p values refer to the overall comparison between the subgroups?

Answer) p value refer to the ovrall comparison between the 3 groups. We added the phrase “P indicates statistical significance between the 3 groups.” in table 1 and 2.

---

## [Decision Letter · Decision Letter 1]

14 Apr 2021

PONE-D-21-04044R1

Association between serum uric acid and left ventricular hypertrophy/left ventricular diastolic dysfunction in patients with chronic kidney disease

PLOS ONE

Dear Dr. Lee,

Thank you for submitting your manuscript to PLOS ONE. After careful consideration, we feel that it has merit but does not fully meet PLOS ONE’s publication criteria as it currently stands. Therefore, we invite you to submit a revised version of the manuscript that addresses the points raised during the review process.

We look forward to receiving your revised manuscript.

Kind regards,

Hans-Peter Brunner-La Rocca, M.D.

Academic Editor

PLOS ONE

Journal Requirements:

Reviewers' comments:

Reviewer's Responses to Questions

**Comments to the Author**

1. If the authors have adequately addressed your comments raised in a previous round of review and you feel that this manuscript is now acceptable for publication, you may indicate that here to bypass the “Comments to the Author” section, enter your conflict of interest statement in the “Confidential to Editor” section, and submit your "Accept" recommendation.

Reviewer #1: (No Response)

Reviewer #2: All comments have been addressed

2. Is the manuscript technically sound, and do the data support the conclusions?

Reviewer #1: Yes

Reviewer #2: Yes

3. Has the statistical analysis been performed appropriately and rigorously? 

Reviewer #1: Yes

Reviewer #2: Yes

4. Have the authors made all data underlying the findings in their manuscript fully available?

Reviewer #1: Yes

Reviewer #2: Yes

5. Is the manuscript presented in an intelligible fashion and written in standard English?

Reviewer #1: Yes

Reviewer #2: Yes

6. Review Comments to the Author

Reviewer #1: Dear Authors,

Thank you for resubmitting your study addressing the recommendations/suggestions made in the former review.

I consider that all the issues asked by me were properly answered, except for the following:

Line 246: “... CI: 0.812–0.860), which was significant higher than that of SUA alone (0.836 vs. 0.803, P < 0.001).”

The correct is “....significantly higher than....”

Congratulations and my best regards

Reviewer #2: Thank you for answering all the comments and queries. Thank you for this nice manuscript. I have no further comments

7. PLOS authors have the option to publish the peer review history of their article (what does this mean?). If published, this will include your full peer review and any attached files.

Reviewer #1: No

Reviewer #2: **Yes: **Konstantinos Papadopoulos

---

## [Author Response · Author response to Decision Letter 1]

14 Apr 2021

PONE-D-21-04044R1

Association between serum uric acid and left ventricular hypertrophy/left ventricular diastolic dysfunction in patients with chronic kidney disease

Reviewer #1: Dear Authors

Thank you for resubmitting your study addressing the recommendations/suggestions made in the former review.

I consider that all the issues asked by me were properly answered, except for the following:

Line 246: “... CI: 0.812–0.860), which was significant higher than that of SUA alone (0.836 vs. 0.803, P < 0.001).”

The correct is “....significantly higher than....”

Answer) As you commented, we corrected “significant” to “significantly”

---

## [Editor Report · Decision Letter 2]

26 Apr 2021

Association between serum uric acid and left ventricular hypertrophy/left ventricular diastolic dysfunction in patients with chronic kidney disease

PONE-D-21-04044R2

Dear Dr. Lee,

We’re pleased to inform you that your manuscript has been judged scientifically suitable for publication and will be formally accepted for publication once it meets all outstanding technical requirements.

Kind regards,

Hans-Peter Brunner-La Rocca, M.D.

Academic Editor

PLOS ONE
---

## [Editor Report · Acceptance letter]

28 Apr 2021

PONE-D-21-04044R2 

Association between serum uric acid and left ventricular hypertrophy/left ventricular diastolic dysfunction in patients with chronic kidney disease 

Dear Dr. Lee:

I'm pleased to inform you that your manuscript has been deemed suitable for publication in PLOS ONE. Congratulations! Your manuscript is now with our production department. 

Kind regards, 

on behalf of

Dr. Hans-Peter Brunner-La Rocca 

Academic Editor

PLOS ONE